# The Effect of Traumatic Brain Injury on Sleep Architecture and Circadian Rhythms in Mice—A Comparison of High-Frequency Head Impact and Controlled Cortical Injury

**DOI:** 10.3390/biology11071031

**Published:** 2022-07-08

**Authors:** Holly T. Korthas, Bevan S. Main, Alex C. Harvey, Ruchelle G. Buenaventura, Evan Wicker, Patrick A. Forcelli, Mark P. Burns

**Affiliations:** 1Interdisciplinary Program in Neuroscience, Georgetown University Medical Center, Washington, DC 20057, USA; htk8@georgetown.edu (H.T.K.); paf22@georgetown.edu (P.A.F.); 2Department of Neuroscience, Georgetown University Medical Center, Washington, DC 20057, USA; bsm53@georgetown.edu (B.S.M.); ach140@georgetown.edu (A.C.H.); rgb51@georgetown.edu (R.G.B.); 3Department of Pharmacology and Physiology, Georgetown University Medical Center, Washington, DC 20057, USA; ew521@georgetown.edu

**Keywords:** sleep, circadian rhythm, traumatic brain injury (TBI), mild TBI (mTBI), concussion

## Abstract

**Simple Summary:**

Traumatic brain injury (TBI) is a significant risk factor for the development of sleep and circadian rhythm impairments. In order to understand if TBI models with different injury mechanism, severity and pathology have different sleep and circadian rhythm disruptions, we performed a detailed sleep and circadian analysis of the high-frequency head impact TBI model (a mouse model that mimics sports-related head impacts) and the controlled cortical impact TBI model (a mouse model that mimics severe brain trauma). We found that both TBI models disrupt the ability of brain cells to maintain circadian rhythms; however, both injury groups could still maintain circadian behavior patterns. Both the mild head impact model and the severe brain injury model had normal amount of sleep at 7 d after injury; however, the severe brain injury mice had disrupted brain wave patterns during sleep. We conclude that different types of TBI have different patterns of sleep disruptions.

**Abstract:**

Traumatic brain injury (TBI) is a significant risk factor for the development of sleep and circadian rhythm impairments. In this study we compare the circadian rhythms and sleep patterns in the high-frequency head impact (HFHI) and controlled cortical impact (CCI) mouse models of TBI. These mouse models have different injury mechanisms key differences of pathology in brain regions controlling circadian rhythms and EEG wave generation. We found that both HFHI and CCI caused dysregulation in the diurnal expression of core circadian genes (*Bmal1*, *Clock*, *Per1*,*2*, *Cry1*,*2*) at 24 h post-TBI. CCI mice had reduced locomotor activity on running wheels in the first 7 d post-TBI; however, both CCI and HFHI mice were able to maintain circadian behavior cycles even in the absence of light cues. We used implantable EEG to measure sleep cycles and brain activity and found that there were no differences in the time spent awake, in NREM or REM sleep in either TBI model. However, in the sleep states, CCI mice have reduced delta power in NREM sleep and reduced theta power in REM sleep at 7 d post-TBI. Our data reveal that different types of brain trauma can result in distinct patterns of circadian and sleep disruptions and can be used to better understand the etiology of sleep disorders after TBI.

## 1. Introduction

Traumatic brain injury (TBI) represents a significant public health concern, with an estimated 2.5 million Americans sustaining a TBI every year [1], resulting in short- and long-term physical, emotional and cognitive impairments. Sleep disruptions are frequently reported in TBI, with over 50% of people experiencing some form of sleep disturbance post-injury [2]. These disturbances may present in the form of poor sleep efficiency, shorter sleep duration, greater wake after sleep onset and changes in sleep architecture [3,4]. The nature of these TBI-induced disturbances are heterogenous on multiple levels, with clinical diagnosis of insomnia, obstructive sleep apnea, hypersomnia and narcolepsy reported both acutely and chronically in the recovery phase [5,6,7,8,9,10]. These diagnoses may also vary with injury severity, with mild TBI patients more likely to report an insomnia disorder than moderate or severe TBI patients [11,12], and repetitive TBI is linked to more severe sleep disturbances than a single TBI alone [13]. Disruption of circadian systems may also contribute to symptoms [14], with TBI patients developing delayed sleep phase syndrome [15,16,17], sleep–wake cycle disorder [18] and circadian rhythm disorder after injury [19].

Sleep comprises two distinct states: Rapid-eye movement (REM) sleep and non-REM (NREM) sleep [20]. Healthy sleep cycles through NREM, REM and wakefulness in a consistent pattern, and each of these states has its identifiable EEG patterns [21,22] that shift from more high-frequency bands while awake (alpha, beta, gamma waves) towards more low-frequency bands while asleep (delta and theta waves). States of wakefulness, REM and NREM are identifiable in rodents when electroencephalogram (EEG) signatures are coupled with electromyography (EMG). Wakefulness is identified by low-amplitude, high-frequency EEG waves with consistent EMG activity. REM sleep’s EEG signature is similar to that detected in a “quiet awake” state, with low-amplitude, mixed frequency signature with the presence of theta waves—however, EMG activity is absent. NREM sleep is characterized by high-amplitude, low-frequency EEG waves with no EMG activity [23], with deep sleep containing delta waves and limited theta waves. Sleep states can be determined by examining the relative activity of the different frequency bands, while the power of each brain wave represents the amount of activity in the specific frequency band.

Although sleep–wake disturbances after TBI have long been recognized in humans, the mechanisms underlying them are still unclear. It is only in recent times that researchers have begun to explore the pathophysiology that underlies sleep related disturbances, utilizing animal models of TBI [24]. However, similar to human studies, heterogeneity is a key barrier in our understanding of sleep and circadian disruptions in animal models. In severe TBI models there is evidence of acute and chronic sleep problems in rodents. Controlled cortical impact (CCI) mice demonstrate sleep–wake disturbances acutely and chronically [25,26]. In lateral fluid percussion injury (FPI), actigraphy monitoring showed decreased dark phase (mouse active phase) activity in the acute to sub-acute phase post injury [27]. Studies using piezoelectric cage systems expanded on these findings, with alterations in post-traumatic sleep in the first week after FPI, but not beyond [28,29,30]. In contrast, studies in closed head injury models have variable results with changes in NREM sleep [31], shorter bouts of wakefulness [32] or no differences [33] being reported.

The reasons for the variability in sleep disturbances in humans may be due to injury heterogeneity in the patient population. Patients are not striated by injury mechanism, and the damaged brain region is not considered in most study designs. As such, there are limited data as to the association of injury type with sleep/circadian disorders. The inherent power of our current pre-clinical models of TBI is their ability to display phenotypes that may aid us to understand mechanistic translations to the human condition. It is clear that deeper characterization of animal models of TBI are required in order to gain a greater understanding of the pathophysiology of sleep and circadian-related disturbances in TBI. In this study, we sought to thoroughly examine sleep and circadian disruptions in two models of TBI that have different mechanisms of injury and different pathology presentation.

In this study, we are using a repetitive closed-head, mild impact model and comparing it to a moderate/severe open head controlled cortical contusion impact model. The high-frequency head impact (HFH) and the controlled cortical impact (CCI) models have some similarities such as chronic cognitive dysfunction—however, they are not similar in terms of their pathology [34,35]. The CCI model is a single severe contusion TBI model that has been widely characterized. It presents with cortical tissue loss, hippocampal atrophy, neuronal cell death and widespread neuroinflammation. Neuroinflammation in the CCI rodent is not limited to the site of injury, but can be found remote from the injury site in areas important to the sleep cycle and arousal, including the thalamus [36] and hypothalamus [25]. In contrast, the HFHI model is a model of high-frequency, highly repetitive but very mild head impacts that does not result in tissue loss, neuronal cell death or widespread neuroinflammation [35]. It does present with specific physiological abnormalities such as chronic changes to glutamate receptors [35] and chronic inflammatory pathology in the optic tract that could affect circadian entrainment [35,37]. Rotational injury in closed head models with an unrestrained skull can also result in damage to the brainstem [38], which may affect the ascending reticular activating system that is important for sleep–wake transitions.

In order to characterize, compare and contrast the sleep patterns of HFHI and CCI models, we used EEG to measure all facets of sleep–wake disturbance, including sleep architecture and power analysis of brain waves post-injury. Additionally, given that circadian biology plays a key role in regulating sleep–wake cycles, we assessed sleep cycles and circadian behavior and assessed changes in the diurnal expression of circadian genes after injury.

## 2. Materials and Methods


Animals


Male C57Bl/6J mice were purchased from Jackson Laboratories (Bar Harbor, ME, USA), and were aged between 3–5 months old at time of injury (24–32 g). Separate cohorts of mice were used for each experiment. Unless stated, mice were housed 5 per cage under 12:12 h light:dark cycles with food and water available ad libitum, and maintained at a temperature of 18–24 °C and 40–60% humidity. All animal procedures outlined were performed in accordance with protocols approved by the Georgetown University Animal Care and Use Committee. As females are reported to have higher rates of sleep disturbances after TBI [39], this study is limited in that the conclusions are not generalizable to the female population.


High-frequency Head Impact (HF-HI) injury model


Mice were anesthetized in a chamber with 3% isoflurane (1.5 L/min oxygen) for 2 min, then placed in the injury device with 3% isoflurane delivered through a nosecone for an additional 1 min of isoflurane (for a total of 3 min under isoflurane). Unrestrained head impact occurred using a 10 mm diameter Teflon-tipped pneumatic impactor at 2.35 m/s with an impact depth of 7.5 mm impact depth with the deceleration of head deflection and rotation being absorbed by a gel pad. We delivered 5 consecutive impacts each day within a 10 s period, with the protocol repeated for 6 consecutive days (total of 30 head impacts). Sham mice received the same amount of anesthesia and handling, but without the impacts to the head. The behavioral, pathological and physiological changes that occur in the HFHI model have been previously described in detail [40,41].


Controlled Cortical Impact (CCI) injury model


Mice were anesthetized with isoflurane (induction at 4% in 1.5 L/min oxygen) and placed on a custom-made stereotaxic frame with built-in heating bed to maintain body temperature at 37 °C for the duration of the surgery. The mouse’s head was secured in the stereotaxic frame, bupivacaine was injected at the surgical site, then clipped and sanitized with iodine and ethanol pads. A 10 mm midline scalp incision was made, followed by a 4 mm craniotomy over the center of the left parietal bone. The 3.5 mm steel impactor tip of a Leica Impact One Stereotaxic Impactor device was slowly lowered to contact the exposed dura at the craniotomy site, then the top was retracted and set to the desired 1.5 mm injury depth before inducing impact. The impact occurred with a velocity of 5.25 m/s, a 100 ms dwell time and a 1.5 mm injury depth. After the injury, the incision was closed using wound clips, the animal was removed from anesthesia and placed on a heating pad for recovery. Sham injury consisted of anesthesia and the midline incision, with no craniotomy or impact. Wound clips were removed 10–14 days post-surgery [40,42].


EEG/EMG Implantation


Bipotential HD-X02 EEG/EMG implants with the PhysioTel Telemetry System (Data Science International, St. Paul, MN, USA) were used to assess EEG parameters and overall sleep architecture. Mice were anesthetized with isoflurane and their heads secured in a stereotactic device with body temperature maintained by heating pad. The surgical site was clipped and sanitized with iodine and ethanol and a 20–30 mm midline incision was made through the scalp. Two EMG leads were implanted in parallel in the cervical trapezius muscle. For the EEG leads, two 1.0 mm bore holes were drilled through the left frontal and the right parietal skull, being careful not to perforate the dura. Lead wires were inserted into the bore holes to contact the dura and secured with dental cement. The incision was closed using non-absorbable sutures. The telemetry device was housed in a subcutaneous space on the left flank. SR-buprenorphine was administered as post-surgical analgesia and triple antibiotic ointment as a post-surgical antibiotic. Surgical implantation was done 1-day post-HFHI or simultaneously with CCI surgery. Mice were single housed for 1 week of recovery in 12:12 L:D conditions, then EEG/EMG was captured for a 24 h period and analyzed using LabChart software v8.12 (AD Instruments, Colorado Spring, CO, USA).


Circadian Wheel-Running Behavior


For circadian behavior, mice were singly housed in separate cages (35.3 × 23.5 × 20 cm) with running wheels (12.7 cm diameter, Lafayette Instrument Model 80820) and ad libitum access to food and water. Wheel running activity was recorded using Scurry Activity Monitoring Software (Lafayette Instruments, Lafayette, IN, USA) for 7 days on a 12:12 LD cycle (lights on at 6 a.m.), then for 15 days in total darkness conditions (DD), then 7 days on a 12:12 LD cycle (lights on at 6 a.m.) for re-entrainment. For analysis of circadian activity, *Clock*Lab Analysis Software Version 6 (Actimetrics, Wilmette, IL, USA) was used to calculate the period, acrophase, and phase shift for the initial 7-day L:D cycle, the 15 day D:D cycle and the 7 day L:D re-entrainment cycle.


Sleep Architecture


The EEG signal was filtered with a 0.1–50 Hz bandpass filter and the EMG signal was filtered with a 50–200 Hz bandpass filter in LabChart before analyzing sleep architecture using Sirenia Sleep Pro software v2.2.5 (Pinnacle, Lawrence, KS, USA). With Sirenia Sleep Pro, recordings were broken into 10 s epochs and coded as wake, NREM or REM using the automatic cluster scoring tool followed by manual scoring. Wakefulness was categorized by low-amplitude, high-frequency EEG waves with EMG activity. NREM was categorized as high amplitude, low-frequency EEG waves with no EMG activity. REM was characterized by low-amplitude, high-frequency EEG waves with no EMG activity. The number of minutes and the average bout length of each behavior state over the 12 h light cycle and the 12 h dark cycle were assessed, as was the power of the delta (0.5–4 Hz), theta (5.5–8.5 Hz), alpha (8–13 Hz), beta (13–30 Hz) and low gamma (35–44 Hz) frequency bands.


Circadian Gene Analysis


Mice were exposed to HF-HI or sham injury between ZT0 (6 a.m.)–ZT5 (11 a.m.) each day. For CCI, mice were exposed to sham or injury surgery between ZT3 (9 a.m.) and ZT6 (12 p.m.). At ZT0 on the day after the final injury day, a subset of each group of mice was euthanized by cardiac perfusion with PBS every 4 h for 24 h. For overnight data collection time points, mice were euthanized in a dark room under red light. RNA was extracted from the hypothalamus, brainstem, and cortex with TRIzol (15596026, Invitrogen, Waltham, MA, USA). The purity and concentration of the extracted RNA were determined using a NanoDrop 1000 spectrophotometer (Thermo Fisher Scientific, Waltham, MA, USA), and 1 μg of RNA was reverse transcribed into complementary DNA (cDNA) with a High-Capacity RNA-to-cDNA Reverse Transcription Kit (4368814, Applied Biosystems, Waltham, MA, USA). cDNA was diluted 1:3 using diethylpyrocarbonate-treated water before being used in RT-QPCR. RT-QPCR was quantified in triplicate using 384-well plates (4309849, Applied Biosystems) using the Prism 7900HT fast sequency detection system (Applied Biosystems). Power SYBR^®^ Green PCR Master Mix (4367659, Applied Biosystems) was used with SYBR probes and analyzed under the following cycle conditions: 50 °C for 2 min, 95 °C for 2 min, (95 °C for 15 s, 60 °C for 15 s, 72 °C for 15 s) for 40 cycles, 95 °C for 15 s for dissociation curve. SDS 2.4 software (Applied Biosystems) was used to generate threshold cycle values (Ct), and fold change in MRNA expression was calculated with the ΔΔCt method (2^−ΔΔCt^). GAPDH was used as a housekeeping gene, and fold change was normalized to each group’s own expression at ZT0 (6 a.m.).


Statistics


To analyze the rhythmic expression of the core circadian *Clock* genes, the cosinor method of analysis was used [43], fitting a sine regression curve to the data using the R package cosinor2: Extended Tools for Cosinor Analysis of Rhythms (https://search.r-project.org/CRAN/refmans/cosinor2/html/00Index.html (accessed on 5 March 2022)) to determine the rhythm-adjusted mean of expression (or midline estimating statistic of rhythm (MESOR)) and the timing to the peak in expression during each cycle (acrophase). All other statistical analysis was conducted using GraphPad Prism v9 (GraphPad Software, San Diego, CA, USA). For comparisons between two groups, an unpaired, two-tailed *t*-test was used.

## 3. Results

### 3.1. HFHI and CCI Cause Dysregulation in the Diurnal Expression of Brain Core Circadian Clock Genes

Starting at ZT0 the day following the final impact, we quantified diurnal expression of core circadian clock genes in the brain. We studied the MESOR and acrophase of *Bmal1*, *Clock*, *Per1*, *Per2*, and *Cry2* in multiple brain regions for a 24 h period. The MESOR represents the mean expression level of each gene across the entire time period. The acrophase represents the time of the peak of gene expression.

In the hypothalamus we found that HFHI caused a significant depression of *Bmal1* MESOR (F(_1,8_) = 12.375, *p* = 0.0079), and an advanced acrophase of *Clock* (F(_1,8_) = 6.622, *p* = 0.033) and *Per2* (F(_1,8_) = 8.143, *p* = 0.021) (Figure 1). In the cortex we found that a significant advance occurred in the acrophase of *Bmal1* (F(_1,8_) = 32.0386, *p* = 0.0005), *Per2* (F(_1,8_) = 5.630, *p* = 0.045), and *Cry1* (F(_1,8_) = 9.616, *p* = 0.0146) in HFHI mice (Figure 1). In the brainstem, we found an advance in acrophase in HFHI mice for *Bmal1* (F(_1,7_) = 10.602, *p* = 0.014) and *Per1* (F(_1,7_) = 14.963, *p* = 0.0061), a delay in acrophase in *Cry2* (F(_1,7_) = 37.343, *p* = 0.0005), and increased MESOR for *Bmal1* (F(_1,7_) = 6.71, *p* = 0.0359) (Figure 1).

In the CCI model, we found a significant delay in *Bmal1* (F(_1,6_) = 15.735, *p* = 0.0074), *Clock* (F(_1,6_) = 18.093, *p* = 0.0055), and *Cry2* (F(_1,6_) = 153.608, *p* = 0.0000168) acrophase, and an advance in *Per1* (F(_1,6_) = 7.787, *p* = 0.0316) expression in the hypothalamus (Figure 2). In the ipsilateral cortex, there was a delay in *Bmal1* (F(_1,6_) = 7.314, *p* = 0.035) and *Per2* (F(_1,6_) = 8.176, 0.0288) acrophase, while the acrophase of *Clock* (F(_1,6_) = 711.12, *p* = 0.000000184), *Per1* (F(_1,6_) = 10.13, *p* = 0.019), and *Cry2* (F(_1,6_) = 17.558, *p* = 0.0057) advanced post CCI (Figure 2). The acrophase of all five circadian genes *Bmal1* (F(_1,6_) = 8.771, *p* = 0.0252), *Clock* (F(_1,6_) = 7.0526, *p* = 0.0377), *Per1* (F(_1,6_) = 35.342, *p* = 0.00101), *Per2* (F(_1,6_) = 14.147, *p* = 0.00938), and *Cry2* (F(_1,6_) = 17.558, *p* = 0.0057) advanced in the brainstem post CCI (Figure 2). CCI also caused increases in the MESOR of *Bmal1* (F(_1,6_) = 9.129, *p* = 0.0234) and *Per1* (F(_1,6_) = 16.54, *p* = 0.0066) in brainstem, and *Clock* (F(_1,6_) = 8.986 *p* = 0.0241) in the ipsilateral cortex (Figure 2).

Together these data demonstrate that dysregulation of the diurnal expression of core circadian genes occurs in multiple brain regions, including the key circadian regulatory center, after both repeated head impact and cortical trauma.

### 3.2. HFHI and CCI Do Not Change Circadian Wheel-Running Behavior

To study circadian behavior, we used running wheels to track activity through a month-long light protocol that included 7 days in 12:12 L:D for acclimatization, 15 days in 12:12 D:D to assess the endogenous behavioral rhythm and 7 days in 12:12 L:D for re-entrainment (Figure 3A). This generated an actogram that shows the daily timing of the mouse’s wheel running activity over the course of the month-long light protocol (Figure 3B, representative image).

Both HFHI and sham mice were active from the first day post-injury with 5–6 km of activity. Their activity steadily increased and peaked at over 10 km per 24 h time period. There were no differences in the distance traveled between sham and HFHI in either of the light:dark phases or in the dark:dark phase (Figure 3C,D). We calculated the circadian period, the acrophase and the phase shift of the running wheel activity during the initial 7-day 12:12 L:D acclimatization time, the 15-day free-running 12:12 D:D time, and the final re-entrainment 7-day 12:12 L:D time. We found no significant differences in the circadian period (Figure 3E–G), the acrophase (Figure 3H–J), the phase shift when moving to the dark:dark cycle (Figure 3K) or the number of days to re-entrain form the dark:dark to light:dark cycles (Figure 3L).

We repeated the same experiment design with CCI-injured mice. CCI mice traveled significantly less than sham mice during the first week post-injury (*p* = 0.022) (Figure 4B,C). However, we found no significant differences in the circadian period (Figure 4E–G), the acrophase (Figure 4H–J), the phase shift when moving to the dark:dark cycle (Figure 4K) or the number of days to re-entrain from the dark:dark to light:dark cycles (Figure 4L).

### 3.3. HFHI and CCI Do Not Alter Sleep Macro-Architecture One-Week Post-Injury

In order to elucidate the effect of TBI injury on sleep architecture, mice first underwent the TBI protocols and were then implanted with EEG/EMG telemetry devices. Analysis was performed on a 24 h time period at 7 d post-TBI. EEG/EMG were recorded for 24 h under normal 12:12 L:D conditions (Figure 5A).

Neither HFHI nor CCI mice displayed any differences in total time spent awake, in NREM or in REM sleep during either the light cycle (6 a.m.–6 p.m.) or dark cycle (6 p.m.–6 a.m.) (Figure 5B,C).

We did observe a 31% reduction in the average REM bout length during the dark cycle in HFHI mice (*p* = 0.048) but no changes in the time spent in any behavior state in the dark cycle (Figure 5D). In CCI mice, we found no differences in the average bout length of time spent awake, in NREM or in REM sleep in either the light or dark cycle (Figure 5E).

### 3.4. Sleep Micro-Architecture Is Changed in CCI, but Not Hfhi Mice

To further explore sleep architecture following TBI, we quantified power of specific frequency bands in the NREM and REM stages of sleep during the light and dark stages in CCI and HFHI mice. Following HFHI, there were no differences in the spectral power of the delta frequency band (Figure 6A) or the theta frequency band (Figure 6B). However, we found that CCI mice had a significant decrease in the spectral power for low-frequency delta and theta waves during the sleep stages. In NREM sleep in the dark phase (6 p.m.–6 a.m.) we found that CCI mice displayed a 55% reduction in delta power (*p* = 0.0127) (Figure 6B). We also found a 59% reduction in theta power during REM sleep during the dark phase (*p* = 0.032) (Figure 6D). The average power for all EEG frequency bands are available in Appendix A.

## 4. Discussion

In this study we compared the sleep and circadian profiles of two distinct mouse models of TBI—a closed head repeat impact model, and a severe cortical impact model. For the first time, we report the diurnal expression of core circadian genes across a 24 h period after experimental TBI and find that both the head impact model and severe TBI model disrupt the natural cycle of these pacemaker genes in multiple brain regions. We also report distinct changes in the sleep architecture after controlled cortical impact TBI, specifically in NREM and REM sleep, that could impact sleep quality, memory consolidation and behavior after TBI.

We found that both HFHI and CCI alter diurnal expression of genes involved in circadian rhythm control. The bidirectional control of the circadian clock is very tightly controlled, with Bmal1 and Clock protein required to be produced simultaneously so it can dimerize to allow the production of Per and Cry proteins, which then dimerize to switch off the circadian clock and start the cycle again. Any dysregulation of the production of Bmal1 with Clock or Per with Cry will cause a significant failure of the circadian clock. In all brain regions studied, we found that the circadian clock was dysregulated, indicating an out-of-sync cellular clock in the brain. We found this dysregulation in both the closed head impact and the cortical impact injury with both HFHI and CCI causing timing shifts across multiple genes in multiple brain regions, desynchronization of the timing of the co-expressed genes and changes to mean *Bmal1* expression levels. Diurnal shifts in expression have previously been reported in the suprachiasmatic nucleus of LFP rats [44]; however, TBI studies have often focused only on single timepoint expression changes [45], which gives little clarity on how either the MESOR or acrophase of the circadian cycle is impacted by TBI. The mechanism that underlies these circadian gene changes throughout the brain is unknown; however, there are multiple different mechanisms that could affect circadian rhythms in CCI and HFHI mice. Circadian rhythms are initially entrained by the transmission of light from photosensitive retinal ganglion cells to the suprachiasmatic nuclei through the optic nerve. Damage to the optic nerve has the potential to disrupt this entrainment through either direct damage to the retinohypothalamic tract or abnormal and mistimed firing of retinohypothalamic neurons. Like most repetitive mTBI models, the repeat impacts in HFHI mice causes chronic optic nerve/tract inflammation and damage [37,46], which could play a role in the disrupted acrophase in the hypothalamus and other brain regions. In contrast, CCI mice do not have optic nerve or tract damage; however, inflammation has been reported in the ipsilateral hypothalamus of CCI mice [25], and this could disrupt the circadian cycle in CCI mice. In addition to inflammation, circadian rhythms are modulated by glutamate signaling, with glutamate release and uptake by astrocytes being regulated in a circadian fashion [47,48]. HFHI mice have global impairments in glutamatergic synaptic signaling following repeated head impact [35], and the CCI mice have well-documented changes to glutamatergic signaling [49], changes to excitatory synapse density and structure [50,51], chronic astrogliosis [52] and reductions in astrocytic glutamate transporters [53]. In both our TBI models, these disruptions to glutamate cycles and signaling could contribute to the misfiring of circadian rhythms and consequential acrophase shifts throughout the brain.

In our models, the change in circadian genes did not result in a shift in running wheel circadian behavior in either of our TBI models. We used voluntary running wheels to track activity and observed the role of light entrainment on locomotor activity—which commenced as soon as the lights were turned off. This entrainment was observed in sham, HFHI and CCI mice. In all groups, switching to a 2-week 24 h dark period caused a clear drift in the timing of activity onset—however, this drift was similar across groups, and we did not discern any significant differences between TBI and controls. All groups were similarly able to re-entrain when we reverted to a 12 h:12 h L:D cycle. The key question is why do we not observe changes in locomotor activity rhythms when the brain’s cellular clock is disrupted by TBI? The answer is that the circadian control of behavior is regulated by many different inputs—and not just by the brain’s cellular clock. Circadian behavior can be regulated by many peripheral inputs, including light entrainment, feeding patterns, noise and smell. This is best seen in *Clock*-deficient mice that lack the ability to make the key Clock regulatory protein. Clock-deficient mice can still entrain their behavior to the light and still maintain a circadian rhythmicity in a constant dark:dark cycle [54].

This is not to say that we do not observe any difference in locomotor activity in TBI mice. CCI mice, but not HFHI mice, have decreased locomotor activity in the week following injury that recovers over time. This is similar to previous reports of reduced activity in FPI mice the day immediately after surgery [44], but not at 2 d post-injury. Weight-drop closed head injury mice do show a shift in circadian behavior with single mTBI mice (but not repeat mTBI mice) showing reduced activity in the dark cycle for the first 72 h post-injury. In contrast, repeat mTBI mice (but not single mTBI mice) have increased activity in the light cycle for the first 72 h post-injury [45]. It is also worth noting that the use of enriched environments, exercise and BDNF are known to improve outcomes after TBI [55,56], and we cannot rule out that our use of stimulating environments and the ability to run on average 9–10 km every 24 h may have improved brain health and reduced our ability to detect circadian changes after TBI.

The etiology of sleep disturbances after TBI are not fully understood. Sleep disruptions can develop in either the acute or chronic phase, and inflammation has been implicated as a key factor that might drive the onset of sleep disturbances. Many models of TBI show neuron cell death, axonal damage and inflammation in the brainstem, thalamus and hypothalamus [25,36,38]—critical brain regions that house sleep centers and arousal centers. Given the timeframe at which we found differences in circadian genes (1 d) and locomotor activity (1–7 d) post-TBI, we wanted to focus our sleep study on this time window. Given that the mice required a recovery period from the invasive EEG implant surgery, the earliest timepoint we could study was at 7 d post-injury. In our study, we found no change to the microarchitecture of sleep in either CCI or HFHI mice. Other studies report discrete sleep changes after TBI. In the acute phase post-injury (1–2 d), a mild weight-drop model found no changes in gross sleep architecture or the timing of nocturnal sleep/wake behavior cycles, but reported fewer “long-wake” behavior bouts after injury [32]. Other studies in weight drop models reported no change at this timepoint [31]. In the subacute injury phase (5–10 d post-TBI), LFP mice have increased sleep time and longer bouts of sleep compared to sham when measured using a piezoelectric sleep system [30]; however, these sleep disruptions resolved by 2 weeks post-injury [30], and other studies reported no differences in sleep phases at this time in the weight drop model [31]. Chronic studies (>14 d post-TBI) show that sleep disturbances can develop over time, with LFP-rats developing NREM and NREM fragmentation at 29 d, but not at 6 d or 19 d post-injury [57], but again no changes were found in REM or NREM in the weight-drop model [31]. In another closed head injury model, the hit-and-run mouse model of repeat mTBI, sleep became fragmented, with injured mice spending more time awake and less time in NREM sleep at 1 m post-injury [31]. These heterogenous data indicate that more studies are needed in TBI animal models to clarify the role of injury type and time on sleep pathology.

The use of EEG to study the sleep microarchitecture allows for a deeper understanding of physiological changes in brain activity that might underlie sleep dysfunction. NREM sleep is characterized by the appearance of deep, slow delta waves and is a period of sleep when the brain consolidates memories. We found that delta power was decreased in NREM sleep in CCI mice, but not HFHI mice, with a >60% decrease in power compared to sham mice. REM sleep is characterized by the presence of theta waves, and we found a reduction in theta waves in CCI, but not HFHI mice. There are only limited sleep studies of rodent TBI models with which to compare our results. Studies in a weight-drop mouse model found that delta waves were increased in the awake state at 2 d, but not 1 d post TBI [32]. In the LFP injury mouse model, it was shown that mice had higher theta:alpha ratios at 8–13 days post-injury, as well as changes to slow waves in wakefulness, which the authors interpreted as a greater homeostatic drive for sleep after TBI [58]. In chronic studies the hit-and-run mice demonstrate a shift to a higher average frequency in the delta frequency band compared to shams [31]. In contrast, in human TBI studies there is no clear coherence, with studies showing no difference [59], an increase in low power bands during NREM sleep [60,61] or a decrease in low power bands during NREM sleep [4,62]. How TBI alters EEG patterns is poorly understood, but damage to, and chronic inflammation of, the origination centers of this electrical activity is a plausible explanation for EEG variation after injury. For example, delta waves can originate in either the cortex or the thalamus. Delta waves from the cortex are regulated by the suprachiasmatic nucleus of the hypothalamus, which is affected by neuroinflammation in the CCI model [25]. The CCI model also has widespread inflammation in the thalamus [36], which could interrupt the generation of thalamic delta waves under the control of the reticular formation. Unlike the CCI model, delta waves are unaffected in HFHI mice, where inflammation of the thalamus does not occur.

## 5. Conclusions

In this study, we compared and contrasted two mouse models of TBI that have different mechanisms of injury and different neuropathology. We reported similarities in the ability of both types of injury to disrupt the expression of circadian genes in the brain; however, these changes to the brain’s cellular rhythms do not translate to a shift in circadian locomotor patterns. We also reported that despite a lack of gross sleep architecture changes, the CCI mouse model presents with decreased delta wave activity in NREM sleep and decreased theta wave activity in REM sleep at 1 week post injury. Our data demonstrate that different TBI models have different effects on sleep and circadian rhythm patterns after TBI and that model selection is an important factor when designing sleep studies for pre-clinical TBI.

## Figures and Tables

**Figure 1 biology-11-01031-f001:**
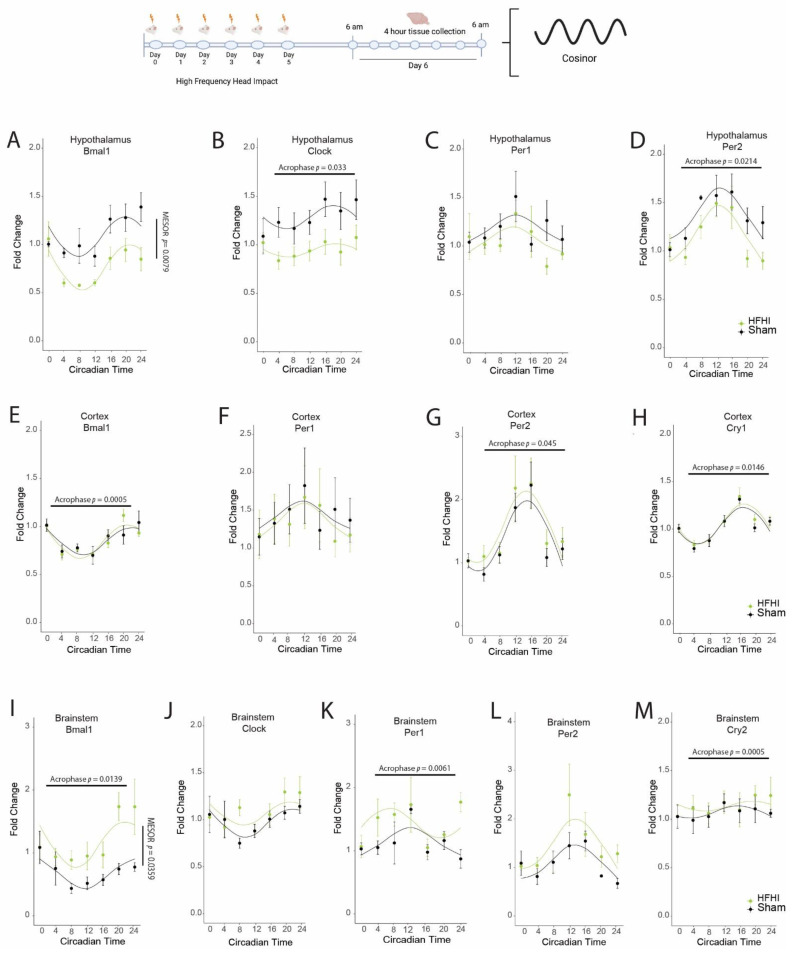
HFHI alters diurnal expression of core circadian genes in multiple brain regions. Schematic of the experimental design showing that mice were exposed to HFHI or sham procedures. Starting at ZT0 the following day, brain tissue was collected every 4 h for 24 h. In the hypothalamus we quantified (**A**) *Bmal1*, (**B**) *Clock*, (**C**) *Per1* & (**D**) *Per2* mRNA. In the cortex we quantified (**E**) *Bmal1*, (**F**) *Per1*, (**G**) *Per2* and (**H**) *Cry2* mRNA. In the brainstem we quantified (**I**) *Bmal1*, (**J**) *Clock*, (**K**) *Per1*, (**L**) *Per2* and (**M**) *Cry2* mRNA. Changes to MESOR or acrophase are indicated on the individual graphs. Each datapoint represents the mean ± SEM of *n* = 4–5.

**Figure 2 biology-11-01031-f002:**
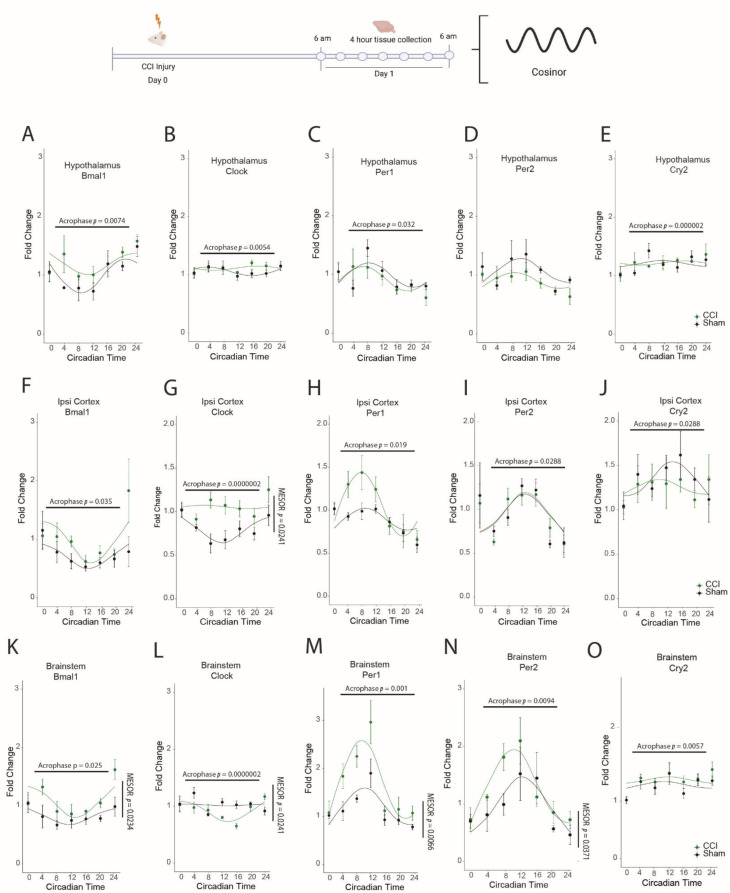
CCI alters diurnal expression of core circadian genes in multiple brain regions. Schematic of the experimental design showing that mice were exposed to CCI or sham surgery. Starting at ZT0 the following day, brain tissue was collected every 4 h for 24 h. In the hypothalamus we quantified (**A**) *Bmal1*, (**B**) *Clock*, (**C**) *Per1*, (**D**) *Per2* and (**E**) *Cry2*. In the ipsilateral cortex we quantified (**F**) *Bmal1*, (**G**) *Clock*, (**H**) *Per1*, (**I**) *Per2* and (**J**) *Cry2*. In the brainstem we quantified (**K**) *Bmal1*, (**L**) *Clock*, (**M**) *Per1*, (**N**) *Per2*, and (**O**) *Cry2*. Changes to MESOR or acrophase are indicated on the individual graphs. Each datapoint represents the mean ± SEM of *n* = 4.

**Figure 3 biology-11-01031-f003:**
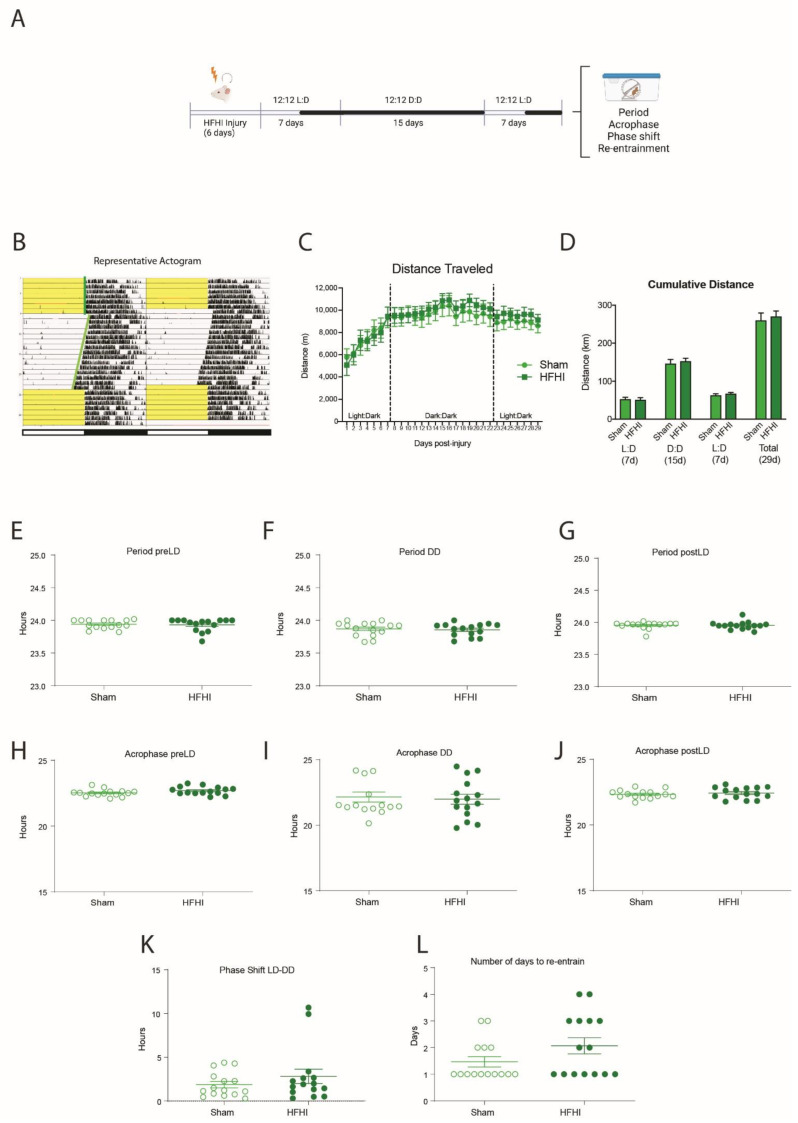
The effect of HFHI on circadian wheel-running behavior. (**A**) Schematic of experimental protocol. Mice were exposed to HFHI or sham protocols for 6 d, then placed in individual running wheel cages the following morning at ZT0. The mice were acclimatized under normal 12 h:12 h light:dark (LD) cycles for 7 d before a 15 d dark:dark (DD) period. Mice were re-entrained to a 12 h:12 h LD cycle for a further 7 d. Spontaneous activity was tracked by running wheel activity. (**B**) Representative image of actogram generated from running wheel activity. Activity counts (in black) are double-plotted over 48 h to see temporal patterns in activity. Yellow background indicates lights on in the behavior room, white background indicates lights were off in the behavior room. Each new 24 h period (right side) is repeated on the line below (left side) to allow for clear visualization of phase shifts. (**C**,**D**) There was no difference in daily distance or cumulative distance traveled between HFHI and sham mice in the first 7 d LD time, the 15 day DD time, or the final 7 day LD time. (**E**–**G**) There was no effect of HFHI on circadian period in the pre-LD, DD or post-LD phases. (**H**–**J**) There was no effect of HFHI on acrophase in the pre-LD, DD or post-LD phases. (**K**) There was no effect of HFHI on phase shift between the pre-LD and the DD phase between injury and sham. (**L**) There was no effect of HFHI on the number of days it took to re-entrain to the post-LD phase. (*n* = 15 per group).

**Figure 4 biology-11-01031-f004:**
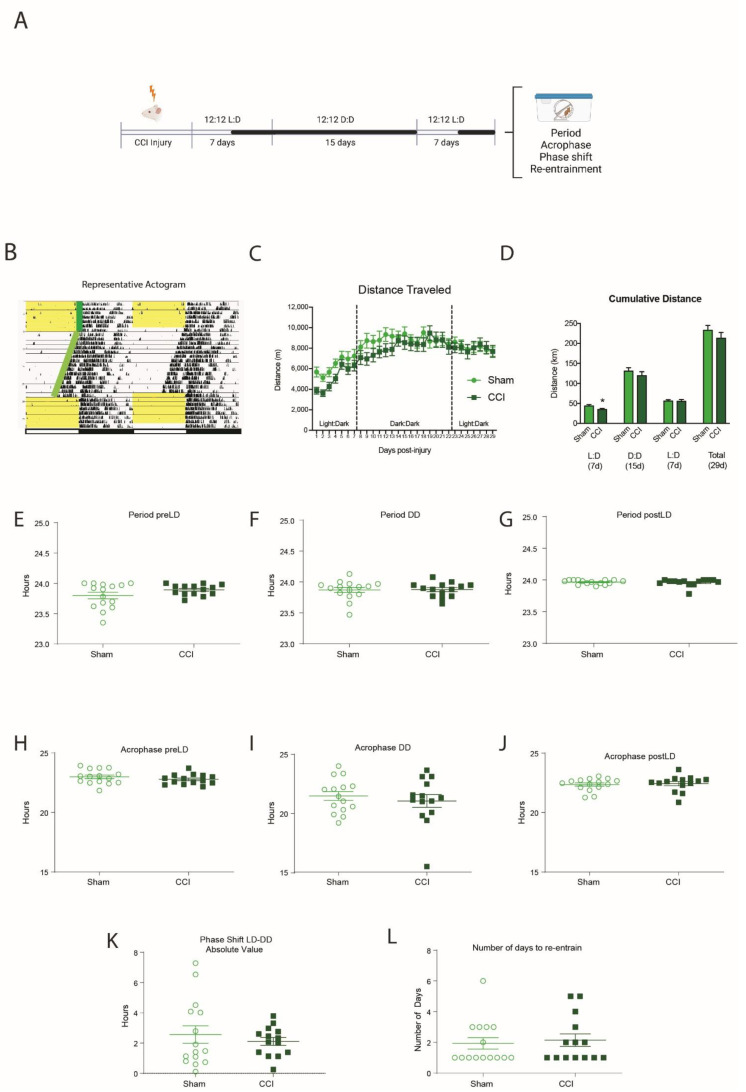
The effect of CCI on circadian wheel-running behavior. (**A**) Schematic of experimental protocol. Mice were exposed to CCI or sham injury, then placed in individual cages the following morning at ZT0. The mice were acclimatized under normal 12 h:12 h light:dark (LD) cycles for 7 d before a 15 d dark:dark (DD) period. Mice were re-entrained to a 12 h:12 h LD cycle for a further 7 d. Spontaneous activity was tracked by running wheel activity. (**B**) Representative image of actogram generated from running wheel activity. Activity counts (in black) are double-plotted over 48 h to see temporal patterns in activity. Yellow background indicates lights on in the behavior room, white background indicates lights were off in the behavior room. Each new 24 h period (right side) is repeated on the line below (left side) to allow for clear visualization of phase shifts. (**C**,**D**) CCI mice ran significantly less than sham during the initial 7 day LD cycle, but reached the same running distances as shams during the 15 day DD and final 7 day LD cycles. (**E**–**G**) There was no effect of CCI on circadian period in the pre-LD, DD or post-LD phases. (**H**–**J**) There was no effect of CCI on acrophase in the pre-LD, DD or post-LD phases. (**K**) There was no effect of CCI on phase shift between the pre-LD and the DD phase between injury and sham. (**L**) There was no effect of CCI on the number of days it took to re-entrain to the post-LD phase. Unpaired *t*-test * *p* = 0.022 (*n* = 14 CCI, 15 sham).

**Figure 5 biology-11-01031-f005:**
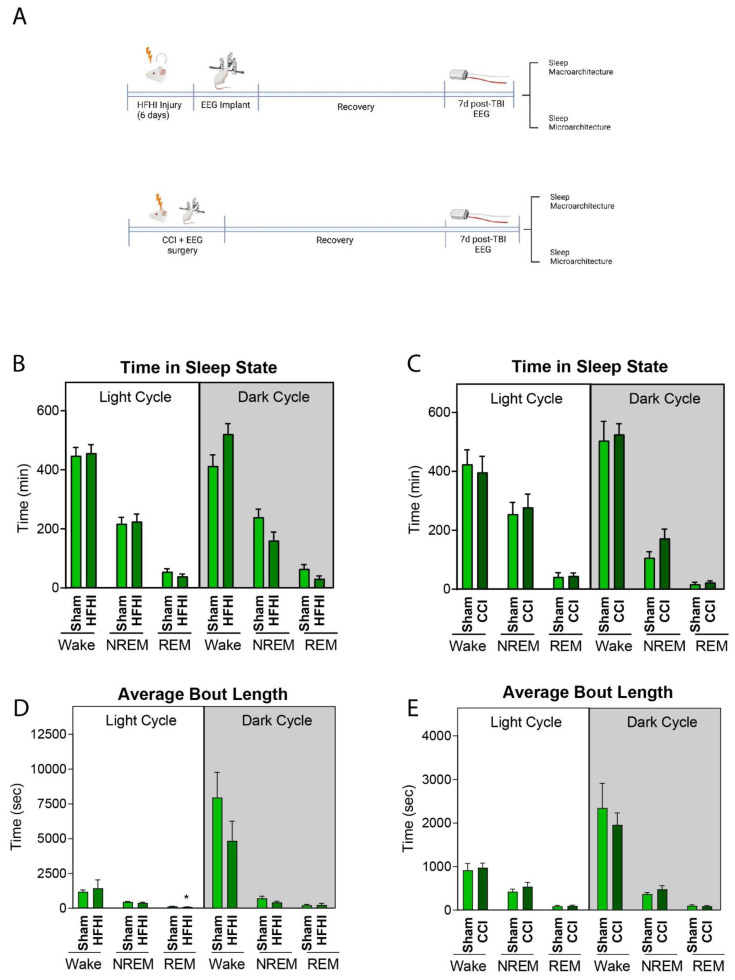
The effect of HFHI and CCI on sleep macro-architecture at 7 d post-injury. (**A**) HFHI and CCI mice were implanted with EEG/EMG telemetry devices and recorded for 24 h in normal 12:12 L:D conditions at 7 d post-injury. (**B**) HFHI caused no change to the amount of time spent in wake, NREM or REM during either the light cycle or the dark cycle. (**C**) CCI caused no difference in the amount of time spent in wake, NREM or REM during either the light cycle or the dark cycle. (**D**) HFHI caused a significant decrease in the bout length of REM during the light cycle compared to sham, but no change in the bout length of wake or NREM. There were no differences in the bout length of wake, NREM or REM during the dark cycle in HFHI mice. (**E**) CCI caused no difference in the average bout length of wake, NREM or REM during either the light cycle or the dark cycle. Unpaired *t*-test, * *p* = 0.048, *n* = 8 sham, 8 HFHI and 8 CCI.

**Figure 6 biology-11-01031-f006:**
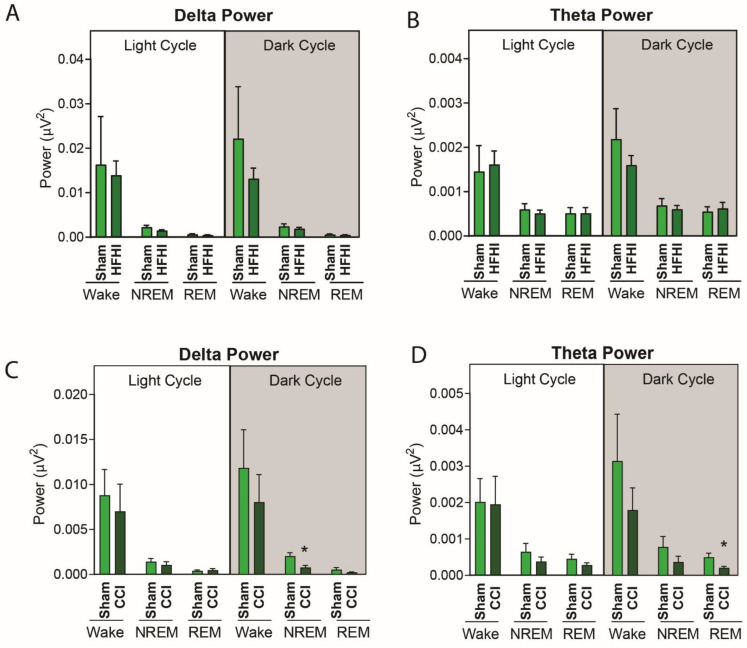
CCI, but not HFHI, alters slow wave sleep at 7 d post-injury. HFHI and CCI mice were implanted with EEG/EMG telemetry devices and recorded for 24 h in normal 12:12 L:D conditions at 7 d post-injury. EEG was analyzed for delta (0.5–4 Hz) and theta (5.5–8.5 Hz) frequency bands, and the power of each frequency band was assessed. Following HFHI, there was no significant change in the amount of (**A**) delta or (**B**) theta band power in either the light cycle or dark cycle. CCI caused a significant decrease in (**C**) delta power during NREM sleep in the dark cycle only and a significant decrease in (**D**) theta power during REM sleep in the dark cycle only. Unpaired *t*-test, * = *p* < 0.05, *n* = 8 sham, 8 HFHI and 8 CCI.

## Data Availability

Data supporting reported results is available by emailing the corresponding author.

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
