# Peer review of "The Effect of Traumatic Brain Injury on Sleep Architecture and Circadian Rhythms in Mice—A Comparison of High-Frequency Head Impact and Controlled Cortical Injury"

_biology, 2022, doi:10.3390/biology11071031_

Round 1
Reviewer 1 Report
Summary: This article provides a robust analysis of immediate, acute (24h post-injury) circadian profiles and the resulting, associated sub-acute (7d post-injury) sleep architecture between two etiologically very different injury models – high frequency, closed head impact (HFHI) and CCI. Altogether, this paper is well written and a pleasure to read. Though there are some concerns in the methodology of these experiments, the comparison between the models offers a unique and necessary analysis of TBI-induced circadian and sleep changes. The last figure and conclusion did present a major concern for the publication. The authors conclude their findings with a hypothesis that microglial inflammation is one mechanism that could mediate these changes in sleep architecture, and the bi-directional relationship of sleep and inflammation in the context of TBI is discussed. However, the authors do not present any direct evidence from this study to support the role of neuroinflammation in mediating sleep or circadian change. It is suggested that the authors revise to address the below listed major and minor concerns to be considered for publication.
Major Concern(s)
- Neuroinflammation hypothesis: Figure 9 and much of the discussion focuses on microglia/neuroinflammation influence on post-TBI sleep but there is no data or evidence of neuroinflammation included in this study. Including some additional brain analysis of microglial reactivity seems possible – assuming the authors collected brain tissue after wheel running and EEG experiments. If this is not possible, then the authors are encouraged to restructure the final figure to summarize the circadian and sleep changes reported in this manuscript. Connecting these changes to previous reports of neuroinflammation after TBI in the text of the discussion would then be more suitable. With no direct microglia or inflammatory data in the manuscript, Figure 9 currently prompts more questions about a lack of data instead of an appreciation of the detailed circadian and sleep analysis that is included in the manuscript.
- Use of non-craniectomy shams in CCI: The authors specify that the CCI shams did not receive a craniectomy page 3 of the methods. Considering a component of this comparison is the difference between a closed head or open head injury model, this naïve rather than true sham-surgery comparison could influence the results. A rationale should be provided for this decision.
- Combining of shams with EEG data: Were there multiple cohorts in which animals from each group (HFHI injury and sham, CCI injury and sham) were represented to allow for pooling and direct comparison between groups? Associated with this, on page 13 in results it is claimed that there was “no difference between HFHI sham and CCI sham” and thus the groups were combined. The clarification of how this was statistically determined, and the results of that analysis should be included in the statistical section of the methods.
- Sleep architecture analysis: On page 4, the authors state that EEG analysis smoothing was performed by requiring behavior states (here assumed to be wake, NREM, or REM) to persist for 6 consecutive 10-second epochs. Several studies have previously shown that phases of mouse sleep, particularly REM, is on average less than a minute long. This provides concern when determining the sleep macro-architecture as the true total time animals spent in NREM/REM could have been overwritten by smoothing. This could particularly affect sleep bout data as a measure of sleep quality. Is it possible that the smoothing analysis was too restrictive in detecting actual differences in REM sleep?
- Were any animals excluded for non-EEG analysis reasons? Additionally, what was the exclusion criteria for outliers in EEG analysis? The individual data points within each experiment group vary from one figure panel to the next (e.g. Figures 6-8, CCI group changes from N = 6 to N = 7 to N = 8 in different panels) – some additional information is needed to appreciate why this is the case.
Minor Concern(s)
- A brief explanation of NREM and REM sleep as well as power bands should be added to the introduction. This will provide more context for readers less familiar with sleep macro- or micro-architecture.
- In the methods on page 2, the authors state that only males were included in this study. There is no further rationale for this decision, which should be addressed. Is there any pre-clinical or clinical evidence to suggest that sex can influence TBI-induced sleep disturbances?
- What was the time between each of the five impacts given on each of the six days in the HFHI model, as described in methods on page 3?
- At the end of page 3 in methods, the authors state that the HD-X02 implant was placed in the “subdural space on the left flank”. Is this a typo and meant to be the sub-cutaneous space?
- On page 4 of methods, there are different times of implantation of the telemetry sensor between CCI and HFHI. Was this time difference accounted for during the week post-implant recovery period? So technically the HFHI animals received 6 days of post-implant recovery while the CCI animals received 7 days? Was this similarly reflected in the shams?
- Gene names throughout the publication should be italicized
- Suggest including a timeline of all experiments completed in Figure 1 – this will help the reader appreciate when circadian and sleep analyses were completed with respect to TBI. Adding some rationale for the specific time points would also be helpful.
- Figure 1 of HFHI circadian gene expression analysis did not include all five genes for all three regions like Figure 2 did for CCI. Even if not significant, it would be nice to see the genes included.
Author Response
The authors would like to thank the reviewers for their thoughtful comments. They have helped strengthen the data presentation, data interpretation and take-home message.
In the resubmission we have addressed all of the comments and below is a point-by-point summary of how we have addressed any reviewers’ concerns. New text in the manuscript is highlighted with a line to the side of the text in the resubmission.
Common Comments by Reviewers:
Reviewer 1
Summary: This article provides a robust analysis of immediate, acute (24h post-injury) circadian profiles and the resulting, associated sub-acute (7d post-injury) sleep architecture between two etiologically very different injury models – high frequency, closed head impact (HFHI) and CCI. Altogether, this paper is well written and a pleasure to read. Though there are some concerns in the methodology of these experiments, the comparison between the models offers a unique and necessary analysis of TBI-induced circadian and sleep changes. The last figure and conclusion did present a major concern for the publication. The authors conclude their findings with a hypothesis that microglial inflammation is one mechanism that could mediate these changes in sleep architecture, and the bi-directional relationship of sleep and inflammation in the context of TBI is discussed. However, the authors do not present any direct evidence from this study to support the role of neuroinflammation in mediating sleep or circadian change. It is suggested that the authors revise to address the below listed major and minor concerns to be considered for publication.
Major Concern(s)
- Neuroinflammation hypothesis: Figure 9 and much of the discussion focuses on microglia/neuroinflammation influence on post-TBI sleep but there is no data or evidence of neuroinflammation included in this study. Including some additional brain analysis of microglial reactivity seems possible – assuming the authors collected brain tissue after wheel running and EEG experiments. If this is not possible, then the authors are encouraged to restructure the final figure to summarize the circadian and sleep changes reported in this manuscript. Connecting these changes to previous reports of neuroinflammation after TBI in the text of the discussion would then be more suitable. With no direct microglia or inflammatory data in the manuscript, Figure 9 currently prompts more questions about a lack of data instead of an appreciation of the detailed circadian and sleep analysis that is included in the manuscript.
- We have removed the summary figure from the manuscript. While we and others have reported on the inflammatory profiles of these mice in detail, we did not add additional quantification in this manuscript as it makes the paper unwieldy. Rather we focus on the effects of two different TBI models on sleep and CR to set the baseline for future mechanistic studies.
- Use of non-craniectomy shams in CCI: The authors specify that the CCI shams did not receive a craniectomy page 3 of the methods. Considering a component of this comparison is the difference between a closed head or open head injury model, this naïve rather than true sham-surgery comparison could influence the results. A rationale should be provided for this decision.
- The use of craniotomy as a control for CCI remains an area of controversy in preclinical research. The craniotomy alone can cause inflammation, change to ICP, and cell death – essentially it is a mild TBI all by itself. Severe TBI in humans is often combined with craniotomy to reduce ICP, control swelling and prevent death. As such, in our lab we consider the craniotomy part of the CCI injury. Our controls receive the same anesthesia and drugs and skin incisions, but we do not perform craniotomy to prevent injury to the brain. In addition, we feel that any injury to the brain that occurs during craniotomy in CCI surgery is immediately superseded by the impact itself, but the craniotomy then allows for brain swelling to occur and prevents mouse death.
- Combining of shams with EEG data: Were there multiple cohorts in which animals from each group (HFHI injury and sham, CCI injury and sham) were represented to allow for pooling and direct comparison between groups? Associated with this, on page 13 in results it is claimed that there was “no difference between HFHI sham and CCI sham” and thus the groups were combined. The clarification of how this was statistically determined, and the results of that analysis should be included in the statistical section of the methods.
- All of the data presented was collected from different recording sessions staggered over the course of 12 months. To address the combined reviewers’ comments about our initial approach, we have separated out the two experiments (CCI vs HFHI) and kept our analysis separate. The data from the macro-architecture experiments is now shown in Figure 5, the data from the micro-architecture in Figure 6 and the additional information on sleep microarchitecture not shown in the Figure is available in Supplemental Tables 1 and 2.
- Sleep architecture analysis: On page 4, the authors state that EEG analysis smoothing was performed by requiring behavior states (here assumed to be wake, NREM, or REM) to persist for 6 consecutive 10-second epochs. Several studies have previously shown that phases of mouse sleep, particularly REM, is on average less than a minute long. This provides concern when determining the sleep macro-architecture as the true total time animals spent in NREM/REM could have been overwritten by smoothing. This could particularly affect sleep bout data as a measure of sleep quality. Is it possible that the smoothing analysis was too restrictive in detecting actual differences in REM sleep?
- In the resubmission we have reverted our analysis back to raw analysis without smoothing. There remain no differences in the time spent in the awake / NREM / REM states, however we do detect a decrease in REM sleep bout time in the light stage.
- Were any animals excluded for non-EEG analysis reasons? Additionally, what was the exclusion criteria for outliers in EEG analysis? The individual data points within each experiment group vary from one figure panel to the next (e.g. Figures 6-8, CCI group changes from N = 6 to N = 7 to N = 8 in different panels) – some additional information is needed to appreciate why this is the case.
- Initially we used a ROUT analysis to remove outliers to clean our data before final statistics. Given the combined concerns of the reviewers we have reverted to including all mice in the final analysis with no removal of outliers.
Minor Concern(s)
- A brief explanation of NREM and REM sleep as well as power bands should be added to the introduction. This will provide more context for readers less familiar with sleep macro- or micro-architecture.
- We have included a paragraph on NREM, REM and power in the introduction.
- In the methods on page 2, the authors state that only males were included in this study. There is no further rationale for this decision, which should be addressed. Is there any pre-clinical or clinical evidence to suggest that sex can influence TBI-induced sleep disturbances?
- Recent studies show that females are more susceptible to sleep disturbances compared to males post-TBI (PMID: 31544552). We are limited in our study in that we only used male mice. This reflects the fact that we have only characterized the inflammatory and synaptic changes in our HFHI model in male mice. We have highlighted this limitation in both the methods (animal section).
- What was the time between each of the five impacts given on each of the six days in the HFHI model, as described in methods on page 3?
- The 5 impacts were delivered within a 10s period. This is now described in the methods.
- At the end of page 3 in methods, the authors state that the HD-X02 implant was placed in the “subdural space on the left flank”. Is this a typo and meant to be the sub-cutaneous space?
- Thank you! Apparently we have a brain-centric world view, even with our typos…..
- On page 4 of methods, there are different times of implantation of the telemetry sensor between CCI and HFHI. Was this time difference accounted for during the week post-implant recovery period? So technically the HFHI animals received 6 days of post-implant recovery while the CCI animals received 7 days? Was this similarly reflected in the shams?
- This is correct. The CCI mice and shams had a 7-day recovery period, but the HFHI mice and shams had a 6-day recovery period.
- Gene names throughout the publication should be italicized
- Gene names are now italicized
- Suggest including a timeline of all experiments completed in Figure 1 – this will help the reader appreciate when circadian and sleep analyses were completed with respect to TBI. Adding some rationale for the specific time points would also be helpful.
- We have included timelines in the individual figures.
- Figure 1 of HFHI circadian gene expression analysis did not include all five genes for all three regions like Figure 2 did for CCI. Even if not significant, it would be nice to see the genes included.
- We did initially run all genes, but experienced some technical failures. We no longer have hypothalamus samples remaining to run the missing gene.
Reviewer 2 Report
The authors present data from two different experimental traumatic brain injury models in mice to address an objective that presents sleep outcome differences between the two. There is no clearly articulated hypothesis. Following brain injury, but not before, outcomes include temporal and regional expression of sleep genes, wheel-running circadian behavior, and EMG/EEG sleep state recording. The studies may have been conducted with rigor, however the reporting of methods and results have limited rigor in their presentation. The authors present findings on changes in gene expression, no change in wheel running, and inconsistent outcomes on EEG sleep states. The manuscript lacks a cohesive story to convey the impact of the results.
Overall, the manuscript requires substantial revision for clarity and completeness. Many experimental details are absent or incomplete. Results do not necessarily report statistical results uniformly throughout the document. And, closer attention to detail is required to convey the impact of the results on the field. To this end, the large number of approaches are not necessarily organized or presented in a concise manner to appreciate the manner in which the studies address the hypothesis.
The strength of the manuscript lies in the rigorous collection of samples for circadian expression of genes.
The rationale for the study states that TBI is a “risk factor for the development of chronic sleep and circadian rhythm impairments.” However, none of the outcome measures would qualify has a chronic outcome. The LD/DD experiments are the longest in the manuscript and show no injury effect.
Throughout the manuscript, the authors do not use specific timing. In many instances, there is no mention of the delay or duration for sleep issues. In others, there are vague phrases (e.g., subacute), which are not defined or used systematically throughout the manuscript. There is no indication of study timing in the abstract.
The writing throughout is not scientific. For example, “strong shift”, “very different pathology profiles”, “little EMG activity”, “decrease in power was very apparent“, “strongly altered.” The abundant use of non-quantitative adjectives does not inform the reader of the results.
Last sentence of the introduction requires a revision for clarity.
At the end of the introduction, the purpose of the manuscript does not layout a clear roadmap of the experiments and further does not follow the same sequence of the data.
Add the weight of the mice at injury, particularly because the 12-20 weeks of age is a large discrepancy.
What is the time between high frequency impacts? How were mice in a single cohort cycled through successive impacts?
What does this mean: “surgical site was blocked with bupivacaine”? This procedure is not accurately applied.
Unclear if mice were used in more than one experiment. Were the EEG animals placed in the running wheel cages?
In sleep studies, time in not reported in AM/PM format. Inconsistencies exist throughout the manuscript.
More detail is required with regard to the EEG/EMG smoothing rationale and procedure. Mice are polysomnic species with short sleep bouts (<60 seconds), which would not align with 6 consecutive 10 sec epochs being smoothed. Justify the smoothing approach for the EEG/EMG sleep data.
Timelines for all experiments are critical. The timing between surgeries, injuries, starting sleep recordings, stopping sleep recordings is inconsistent between experiments. More complete timelines would assist the reader to follow the study designs.
Method of euthanasia for the gene analysis mice must be included.
How were RNA purity and concentration determined. The text only indicates the equipment used.
How is fold change abbreviated FT?
The opening of the statistics sections is duplicated from above, without adding additional insight.
Neither MESOR nor acrophase are introduced or explained clearly to allow the reader to follow the impact and meaning of the results.
“In the hypothalamus we found that HFHI caused a significant elevation of Bmal1 MESOR” [Results]. The graphs shows HFHI below the sham levels.
For the gene expression, make the x-axis of all graphs set by 4 hours to match the study design.
According to Fig 3A, the wheel running occurred immediately after the last TBI. This approach does not align with the hypothesis and rationale of chronic sleep disturbances. Further, voluntary wheel running has beneficial effects on outcome. What are the revolutions per hour/day for these animals? Is there an injury effect?
One would expect motor deficits and/or lack of motivation following 30 brain injuries. Where do or would these impairments be detected in the running wheel data?
The results start with rigorous reporting of statistical outcomes (F values, degrees of freedom). These are no longer included in the latter parts of the results.
This reviewer calls into question the duration of sleep bouts recorded. Bout lengths of ~1000 seconds in mice is atypical.
To appreciate the differences between conditions, Figure 5 would benefit from standardizing the y-axes between panels.
For the sleep macro and micro architecture, there are grossly different group sizes (17, 13, 8), which can contribute to spurious statistical findings. Additional detail or random selection is needed to assure that the results are rigorous.
Micrographs are referenced in the figure legend of figure 7 and 8. This is unclear.
Figure 9 legend contains author notes. Histological images are not appropriate for Figure 9, as the studies do not evaluate histology. Figure 9 is conjecture and does not synthesize the data within the manuscript.
This reviewer is not able to accept the last point in the results regarding changes in sleep microarchitecture. The reported results are likely due to the unbalanced study design, large number of statistical tests, or other factors.
In the discussion, what model does 'severe TBI' reference?
The second paragraph of the discussion is more suitable for the introduction in rationalizing the selection of the models.
The discussion of pathology, glutamate, and the suprachiasmatic nucleus are not temporally related to the study design or data. It is unclear how immediate glutamate release would impact sleep microarchitecture at 1 week post-injury.
Causal statements between genes, wheel running, and sleep architecture are inappropriate given the non-causative study design.
“consistent with the existing literature, which has not shown a clear sleep pathology in rodent models.” [Discussion] Discrepancy in the literature cannot explain discrepancy in the data. This statement is out of line.
Editing issues can be found throughout the manuscript. “power bands that could that indicates a hyperarousal” [Discussion] as one example.
Author Response
The authors would like to thank the reviewers for their thoughtful comments. They have helped strengthen the data presentation, data interpretation and take-home message.
In the resubmission we have addressed all of the comments and below is a point-by-point summary of how we have addressed any reviewers’ concerns. New text in the manuscript is highlighted with a line to the side of the text in the resubmission.
Reviewer 2
The authors present data from two different experimental traumatic brain injury models in mice to address an objective that presents sleep outcome differences between the two. There is no clearly articulated hypothesis. Following brain injury, but not before, outcomes include temporal and regional expression of sleep genes, wheel-running circadian behavior, and EMG/EEG sleep state recording. The studies may have been conducted with rigor, however the reporting of methods and results have limited rigor in their presentation. The authors present findings on changes in gene expression, no change in wheel running, and inconsistent outcomes on EEG sleep states. The manuscript lacks a cohesive story to convey the impact of the results.
- The research in this manuscript is not hypothesis driven – instead it attempts to perform a characterization to compare and contrast different injury models at the same time. Both of these injury models display sustained cognitive dysfunction in the acute and chronic phases, but their injury mechanisms and resultant pathology is very different. The goal is to highlight that injury mechanism is important, and should be considered when designing human sleep studies. Furthermore, the data in this manuscript will help sleep and circadian researchers select TBI models that best reflect their research goals, and demonstrate that researchers should characterize their models to ensure they reflect the sleep disturbances they aim to study and treat.
The impact of this research is that the study reinforces the need for more detailed characterization of TBI models. It highlights the fact that sleep microarchitecture can be disrupted, even in the absence of disturbances in sleep microarchitecture. It also demonstrates that the brains cellular circadian clocks can be severely disrupted after TBI, but this may not be apparent in the TBI subject given that the ability of the body to maintain behavioral circadian patterns remains unaffected. Overall these data bring to light a number of clinically translational points that are important to the TBI field as we struggle to understand sleep and circadian disruptions after TBI.
Overall, the manuscript requires substantial revision for clarity and completeness. Many experimental details are absent or incomplete. Results do not necessarily report statistical results uniformly throughout the document. And, closer attention to detail is required to convey the impact of the results on the field. To this end, the large number of approaches are not necessarily organized or presented in a concise manner to appreciate the manner in which the studies address the hypothesis.
- We have substantial revised for clarity and completeness. We have reduced the number of figures in the manuscript. For the sleep microarchitecture data, we have now highlighted the changes to low amplitude brain waves that occur during REM and NREM in CCI mice, but included all HFHI and CCI brainwave data in supplemental tables. We have attempted to improve how we convey the impact of our results, and what they mean for the field.
The strength of the manuscript lies in the rigorous collection of samples for circadian expression of genes.
The rationale for the study states that TBI is a “risk factor for the development of chronic sleep and circadian rhythm impairments.” However, none of the outcome measures would qualify has a chronic outcome. The LD/DD experiments are the longest in the manuscript and show no injury effect.
- We have modified the sentence and removed the word “chronic”
Throughout the manuscript, the authors do not use specific timing. In many instances, there is no mention of the delay or duration for sleep issues. In others, there are vague phrases (e.g., subacute), which are not defined or used systematically throughout the manuscript. There is no indication of study timing in the abstract.
- We have added experimental timelines to the figures to make the timing more explicit. We have added more detail in results. The use of acute, subacute and chronic is an attempt to fit our data with existing sleep studies which are reported in a wide variety of times. As originally defined in the manuscript, we used acute to mean 1-2d post injury; subacute for 5-10d post injury and chronic as >14d as these encompass the timepoints of previously published studies that we include in our discussion.
The writing throughout is not scientific. For example, “strong shift”, “very different pathology profiles”, “little EMG activity”, “decrease in power was very apparent“, “strongly altered.” The abundant use of non-quantitative adjectives does not inform the reader of the results.
- We have reorganized some of our discussion to ensure that the description of the different pathology now occurs in the introduction. We have reduced the use of non-quantitative adjectives.
Last sentence of the introduction requires a revision for clarity.
- Introduction has been revised
At the end of the introduction, the purpose of the manuscript does not layout a clear roadmap of the experiments and further does not follow the same sequence of the data.
- Introduction has been revised
Add the weight of the mice at injury, particularly because the 12-20 weeks of age is a large discrepancy.
- Weight range of 24-32g has been included.
What is the time between high frequency impacts? How were mice in a single cohort cycled through successive impacts?
- The 5 daily impacts were delivered within a 10s period. This is now described in the methods.
What does this mean: “surgical site was blocked with bupivacaine”? This procedure is not accurately applied.
- Nerve block. Changed to “bupivacaine was injected at the surgical site”.
Unclear if mice were used in more than one experiment. Were the EEG animals placed in the running wheel cages?
- Separate cohorts of mice were used for each experiment.
In sleep studies, time in not reported in AM/PM format. Inconsistencies exist throughout the manuscript.
- We use light and dark to describe recordings taken at different time periods for the sleep and running wheel studies. We define that 6am-6pm is light and 6pm-6am is dark as is standard in animal facilities. Changing to a time-stamp would not work for the running wheel studies when we have a dark:dark protocol. In addition, using time is difficult for interpretation for the reader as it becomes a three-step interpretive process (time to light status to activity status of the mouse). We do use ZT time for the circadian studies as this is standard for that field.
More detail is required with regard to the EEG/EMG smoothing rationale and procedure. Mice are polysomnic species with short sleep bouts (<60 seconds), which would not align with 6 consecutive 10 sec epochs being smoothed. Justify the smoothing approach for the EEG/EMG sleep data.
- In the resubmission we have reverted our analysis back to raw analysis without smoothing. There remain no differences in the time spent in the awake / NREM / REM states, however we do detect a decrease in REM sleep bout time in the light stage.
Timelines for all experiments are critical. The timing between surgeries, injuries, starting sleep recordings, stopping sleep recordings is inconsistent between experiments. More complete timelines would assist the reader to follow the study designs.
- A timeline has been added to individual figures.
Method of euthanasia for the gene analysis mice must be included.
- Included
How were RNA purity and concentration determined? The text only indicates the equipment used.
- We used the nanodrop equipment to determine the purity of the RNA. We are not adding instructions on the use of the equipment.
How is fold change abbreviated FT?
- Removed
The opening of the statistics sections is duplicated from above, without adding additional insight.
- Removed
Neither MESOR nor acrophase are introduced or explained clearly to allow the reader to follow the impact and meaning of the results.
- We have added the following sentence to the results. “The MESOR represents the mean expression level of each gene across the entire time period. The acrophase represents the time of the peak of gene expression.”
“In the hypothalamus we found that HFHI caused a significant elevation of Bmal1 MESOR” [Results]. The graphs shows HFHI below the sham levels.
- Corrected
For the gene expression, make the x-axis of all graphs set by 4 hours to match the study design.
- The x-axis are all now set to 4h intervals
According to Fig 3A, the wheel running occurred immediately after the last TBI. This approach does not align with the hypothesis and rationale of chronic sleep disturbances. Further, voluntary wheel running has beneficial effects on outcome. What are the revolutions per hour/day for these animals? Is there an injury effect?
- We have removed the word chronic from our initial rationale in response to an earlier reviewer comment. In terms of the running wheel protocol, as the protocol is designed to last a month, we wanted to capture any effect on the acute, subacute and chronic phases – so we put the mice in immediately after the last TBI. We have now included new data to show the effect of TBI on distance travelled in Fig 3C-D and Fig 4C-D. There was no effect of HFHI on distance, but CCI significantly decreased the distance traveled in the first week following injury. This data is also added to our discussion.
One would expect motor deficits and/or lack of motivation following 30 brain injuries. Where do or would these impairments be detected in the running wheel data?
- Given the very mild nature of the HFHI model, our lab would NOT have expected motor deficits after 30 head impacts of the HFHI model. New data is included in Fig 3C-D to show that the HFHI mice travel the same distance each day as the sham mice.
Our lab DOES expect motor deficits after CCI as this is a moderate to severe brain injury, and we have included data in Fig 4C-D to show that there is decreased distance traveled during the first week after CCI. Despite this, we were surprised at how far the CCI mice actually travelled in the first week – they clearly maintain motivation in the first week despite a moderate - severe TBI.
The results start with rigorous reporting of statistical outcomes (F values, degrees of freedom). These are no longer included in the latter parts of the results.
- When we have no significant changes or interactions between groups, we did not include the F or p values for the sake of readability. It made no sense to include detailed statistical analysis for the running wheel and macroarchitecture of sleep as there were no significant differences observed. For the microarchitecture of sleep we included F and p values for the ANOVA in the results, with the posthoc analysis results represented in the figures.
In response to comments from all 3 reviewers we have modified the reporting of the sleep data and separated out the CCI and HFHI experiments. We have only included the slow wave analysis in the figures, but we have included the full data in supplemental tables. This data is now analyzed using an unpaired t-test, and the t- and p-values are reported in Supplemental Tables 1 and 2.
This reviewer calls into question the duration of sleep bouts recorded. Bout lengths of ~1000 seconds in mice is atypical.
- In the resubmission we have used non-smoothed data and reported the results of this analysis. This has reduced the duration of the sleep bouts recorded, and is shown in Fig 5.
To appreciate the differences between conditions, Figure 5 would benefit from standardizing the y-axes between panels.
- Panels in Fig 5 are now grouped in a single graph with a single y-axes
For the sleep macro and micro architecture, there are grossly different group sizes (17, 13, 8), which can contribute to spurious statistical findings. Additional detail or random selection is needed to assure that the results are rigorous.
- For the resubmission we have separated out the sham groups to match their treatment groups. Final group sizes are Sham-HFHI (8), HFHI (12). In a separate experiment we have Sham-CCI (8) and CCI (8). The discrepancy for n in the HFHI study was due to an error with the recording software where only 4 of 8 mice had data captured.
Micrographs are referenced in the figure legend of figure 7 and 8. This is unclear.
- Removed
Figure 9 legend contains author notes. Histological images are not appropriate for Figure 9, as the studies do not evaluate histology. Figure 9 is conjecture and does not synthesize the data within the manuscript.
- In response to reviewer’s comments, we have removed Fig 9
This reviewer is not able to accept the last point in the results regarding changes in sleep microarchitecture. The reported results are likely due to the unbalanced study design, large number of statistical tests, or other factors.
- We have changed how we are reporting the data to reflect these concerns. We don’t have an unbalanced study design, the changes in sleep microarchitecture are only occurring in CCI mice, they are detectable even in the absence of changes to gross sleep, and they are consistent with changes reported in human and mice studies.
In the discussion, what model does 'severe TBI' reference?
- Severe TBI references CCI. In human TBI, injury is striated into mild, moderate and severe TBI. The CCI model is representative of a TBI on the scale of moderate to severe. We call it severe as it creates a large hole in the cortex and hippocampus of the mice, shrinking of the ipsilateral thalamus, and widespread chronic inflammation. This is in contrast to the HFHI model which does not cause cortical, hippocampal or thalamic pathology.
The second paragraph of the discussion is more suitable for the introduction in rationalizing the selection of the models.
- This paragraph is moved to the introduction
The discussion of pathology, glutamate, and the suprachiasmatic nucleus are not temporally related to the study design or data. It is unclear how immediate glutamate release would impact sleep microarchitecture at 1 week post-injury.
- This discussion paragraph is used to speculate on how the different types of TBI might cause problems with circadian rhythms and sleep. It builds upon what we know about pathology in the two mouse models and proposes mechanisms by which each model might have pathology that impacts sleep and circadian rhythms.
- We do not discuss immediate glutamate release, but rather that CCI and HFHI mice have chronic changes to glutamatergic receptors on the synapse that affects their glutamatergic signaling. Given the importance of glutamate signaling to the maintenance of circadian rhythms, this feels like important information.
Causal statements between genes, wheel running, and sleep architecture are inappropriate given the non-causative study design.
- We have removed any causal statements
“consistent with the existing literature, which has not shown a clear sleep pathology in rodent models.” [Discussion] Discrepancy in the literature cannot explain discrepancy in the data. This statement is out of line.
Editing issues can be found throughout the manuscript. “power bands that could that indicates a hyperarousal” [Discussion] as one example.
- Corrected
Reviewer 3 Report
Nice research combining physiology and behavior in two mouse models of TBI. The paper is well written and the content important. I have some questions about the approach to analysis and would like to see a more cohesive discussion tying all of the results together.
The authors describe “smoothing” the sleep architecture data by requiring sleep or wake states to persist for “6 consecutive epochs before changing to another state”. With 10 second epochs this would require a full minute in any given state. I would like to see more rationale for why this was done and why this period of time was chosen, especially given how fragmented typical mouse sleep is. My concern is that this could also bias data from collected TBI animals with more fragmented sleep and wake states. How many epochs are impacted by this criteria?
Sleep micro-architecture analysis is confusing with current reporting language. Unclear what is being reported and what is being compared – are the p values for Tukey or F-test? Why are the degrees of freedom different for delta, theta, and alpha throughout this section?
It's possible that I misunderstand the design (in which case, see comment below re: experimental design) but it seems that the timing of the circadian gene study does not match with either the circadian or sleep behaviors performed (1 day post injury vs. 1 week post injury). What does this mean for the behavioral results?
The discussion is generally well written, but the stories feel disjointed. More cohesive discussion bringing together the circadian and sleep results as well as more summary of how the circadian gene results differ between TBI models (along with discussion interpreting these results). The summary figure at the end is very nice.
Minor:
Meticulous attention to detail when designing and describing the experiments. I found myself a little bit confused about the overall design and how cohorts/experiments were run. A figure or table outlining timing of procedures and sample sizes for each of the experiments would be helpful to include in the methods section.
First line of HFHI and CCI wheel running paragraph on p9 states “to determine if the shifts in circadian genes cause by brain injury can be correlated with shifts in circadian behavior…” is misleading – this is not what was tested in this experiment since 1) different animals and 2) different time points. Please rephrase.
Author Response
The authors would like to thank the reviewers for their thoughtful comments. They have helped strengthen the data presentation, data interpretation and take-home message.
In the resubmission we have addressed all of the comments and below is a point-by-point summary of how we have addressed any reviewers’ concerns. New text in the manuscript is highlighted with a line to the side of the text in the resubmission.
Reviewer 3:
Nice research combining physiology and behavior in two mouse models of TBI. The paper is well written and the content important. I have some questions about the approach to analysis and would like to see a more cohesive discussion tying all of the results together.
The authors describe “smoothing” the sleep architecture data by requiring sleep or wake states to persist for “6 consecutive epochs before changing to another state”. With 10 second epochs this would require a full minute in any given state. I would like to see more rationale for why this was done and why this period of time was chosen, especially given how fragmented typical mouse sleep is. My concern is that this could also bias data from collected TBI animals with more fragmented sleep and wake states. How many epochs are impacted by this criteria?
- In the resubmission we have reverted our analysis back to raw analysis without smoothing. There remain no differences in the time spent in the awake / NREM / REM states, however we do detect a decrease in REM sleep bout time in the light stage.
Sleep micro-architecture analysis is confusing with current reporting language. Unclear what is being reported and what is being compared – are the p values for Tukey or F-test? Why are the degrees of freedom different for delta, theta, and alpha throughout this section?
- Initially we used a ROUT analysis to remove outliers to clean our data before final statistics. This was because we had clear issues with some power bands and we would remove that mouse’s power band from the analysis in all sleep states, but keep their other power bands. Given the concerns of all three reviewers, we have reverted to including all mice in the final analysis with no removal of outliers. We have also separated out the experiments to CCI and HFHI in response to reviewers’ comments. This new study design allows the use of an unpaired t-test, and this data is included in the supplemental tables.
It's possible that I misunderstand the design (in which case, see comment below re: experimental design) but it seems that the timing of the circadian gene study does not match with either the circadian or sleep behaviors performed (1 day post injury vs. 1 week post injury). What does this mean for the behavioral results?
- The circadian studies which initiated this research were performed starting at ZT0 the day after injury and showed that circadian cycles are immediately impacted by head impact and TBI. To follow up on these studies we wanted to examine behavior, and the circadian wheel running is the best-known method to track this. The behavioral activity assay we used requires light training and extended dark:dark periods, so it is not possible to match this one month long protocol to a specific timepoint. We began recording behavior immediately following TBI and in this resubmission we present data showing that running activity is lower in CCI mice compared to sham for the first week after TBI, but normalizes after that time. HFHI mice are not affected and have normal running wheel activity from Day 1.
- The data itself is quite fascinating as it shows a clear disconnect between brain cellular rhythms and behavioral rhythms. Behavioral rhythms are controlled by inputs including light, feeding, gastrointestinal clocks, other peripheral clocks, cellular clocks, sounds, smells etc. Our data shows that even though the brains cellular clocks are severely disrupted in the case of CCI, it does not alter their circadian behavior. This data is seen in other experiments such as Clock knockout mice that have not got a brain cellular clock, but maintain circadian behavior. For the TBI field, this can also mean that TBI patients can have severely disrupted circadian rhythms in the brain that could disrupt hormone production and mental /cognitive function, but might not be apparent in their gross behavioral routines. We have added this discussion to the paper.
- We have added a timeline of the experiments into individual figures to make the experimental design more explicit.
The discussion is generally well written, but the stories feel disjointed. More cohesive discussion bringing together the circadian and sleep results as well as more summary of how the circadian gene results differ between TBI models (along with discussion interpreting these results). The summary figure at the end is very nice.
- Thank you for the suggestions. We have covered a lot of ground in this manuscript, and it has been challenging to write a cohesive manuscript. In this resubmission we have attempted to relate the circadian and sleep studies and added discussion on the circadian disruptions and how they differ in each model. However, given the comments of R1 and R2, we have removed the summary figure.
Minor:
Meticulous attention to detail when designing and describing the experiments. I found myself a little bit confused about the overall design and how cohorts/experiments were run. A figure or table outlining timing of procedures and sample sizes for each of the experiments would be helpful to include in the methods section.
- We have added a timeline of the experiments into individual figures to make the experimental design more explicit.
First line of HFHI and CCI wheel running paragraph on p9 states “to determine if the shifts in circadian genes cause by brain injury can be correlated with shifts in circadian behavior…” is misleading – this is not what was tested in this experiment since 1) different animals and 2) different time points. Please rephrase.
We have removed this line